# Identification and Expression Analysis of Cold Shock Protein 3 (BcCSP3) in Non-Heading Chinese Cabbage (*Brassica rapa* ssp. *chinensis*)

**DOI:** 10.3390/plants9070890

**Published:** 2020-07-14

**Authors:** Feiyi Huang, Jin Wang, Weike Duan, Xilin Hou

**Affiliations:** 1State Key Laboratory of Crop Genetics and Germplasm Enhancement/Key Laboratory of Biology and Germplasm Enhancement of Horticultural Crops in East China, Ministry of Agriculture/Engineering Research Center of Germplasm Enhancement and Utilization of Horticultural Crops, Ministry of Education, Nanjing Agricultural University, Nanjing 210095, China; hfy@njau.edu.cn (F.H.); 2017204023@njau.edu.cn (J.W.); 2015104054@njau.edu.cn (W.D.); 2College of Life Sciences and Food Engineering, Huaiyin Institute of Technology, Huai an 223003, China

**Keywords:** cold shock protein 3, cold stress, expression analysis, subcellular localization, non-heading Chinese cabbage

## Abstract

A cold-related protein, cold shock protein 3 (BcCSP3), was isolated from non-heading Chinese cabbage in this study. BcCSP3 can encode 205 amino acids (aa) with an open reading frame (ORF) of 618 base pairs (bp). Multiple sequence alignment and phylogenetic tree analyses showed that BcCSP3 contains an N-terminal cold shock domain and is highly similar to AtCSP2, their kinship is recent. Real-time quantitative polymerase chain reaction (RT-qPCR) showed that the expression level of *BcCSP3* in stems and leaves is higher than that in roots. Compared with other stress treatments, the change in *BcCSP3* expression level was most pronounced under cold stress. In addition, a *BcCSP3–GFP* fusion protein was localized to the nucleus and cytoplasm. These results indicated that BcCSP3 may play an important role in response to cold stress in non-heading Chinese cabbage. This work may provide a reference for the identification and expression analysis of other *CSP* genes in non-heading Chinese cabbage.

## 1. Introduction

Non-heading Chinese cabbage (*Brassica rapa* ssp. *chinensis*), a widely grown vegetable crop, is also an important *Brassica* crop with exceptional cold resistance [1,2]. As the global climate continues to change, plants (including Pak-choi) must be equipped with appropriate regulatory mechanisms to be able to respond to changes in the surrounding environment [3]. As one of the important environmental factors (abiotic stress), cold stress can affect crop yield and quality by restricting plant growth and development [4]. After a period of low temperature (no freezing), some plants can obtain higher resistance to freezing. This process is called cold acclimation and has been proven by previous studies [5]. Cold acclimation is a complex process with a series of changes in gene expression and protein metabolism [6].

Cold-induced protein, cold shock domain (CSD) protein (CSP), is an evolutionarily conserved nucleic-acid binding domain. Studies have shown these proteins are widely found in animals and plants [7]. Karlson and Imai found that in plants, CSP usually contains an N-terminal CSD region and a glycine-rich region, which is randomly scattered with multiple CCHC zinc fingers (ZnF) [8]. In a prior study, they identified and isolated *WCSP1* from wheat (*Triticum aestivum* L.), which was the first *CSP* isolated from plants [9]. However, previous experiments have shown that WCSP1 mRNA can only be induced by cold and without being regulated by other environmental abiotic stresses (such as salt or ABA stress). These results indicate that WCSP1 induction may be cold-specific [9]. In addition, WCSP1 can bind to DNA and unlock the double-stranded structure of its nucleotides [9,10] and partially supplements the cold sensitivity of the *E. coli* CSP mutant. These findings indicate that WCSP1 has similar functions as *E. coli* CSPs during cold adaptation [11]. The above predecessors’ research on CSP from wheat is more in-depth, but there is relatively little research on CSP from Pak-choi. Based on the above mentioned studies, further research on the cold stress tolerance of Pak-choi could be valuable in improving the growth and development and by extension, the yield and quality of this economically important crop.

To date, four CSD proteins (AtCSP1, AtCSP2, AtCSP3 and AtCSP4) from *Arabidopsis* have been identified and functionally analyzed [12]. Further analysis showed that *AtCSP3* enhanced both the crop cold resistance pathway and C-repeat/dehydration responsive element-binding factor (CBF) pathway, which were two unrelated pathways [13]. *AtCSP2* plays a role as the negative regulator of freezing tolerance and is considered functionally redundant to *AtCSP4* and *AtCSP2*, as it can regulate flowering time and frost resistance of plants compared to *AtCSP3* [14]. It was further found that *AtCSP2* had obvious activity in meristems and developmental tissues of plants, while *AtCSP1* overexpression did not increase the tolerance to low temperature stress [15,16].

Although *CSP* has been identified in many other plants and functional studies of *CSP* have been conducted, information about *CSPs* in Pak-choi (*Brassica rapa* ssp. *chinensis*) is still limited. Moreover, their response mechanism to abiotic stresses is not yet clear. Through sequence alignment and evolutionary tree analysis, we found that BcCSP3 and AtCSP2 proteins are highly similar and have a closer relationship. Therefore, we speculate whether they have similar cold stress emergency response. In this study, we analyzed the protein structure, protein interaction prediction, phylogenetic relationships, conserved motifs and subcellular localization of *BcCSP3*. In addition, the expression level changes of *BcCSP3* in different vegetative organs (roots, stems and leaves) and under multiple abiotic stresses were examined through RT-qPCR test analysis. In all, this work may help to reveal the biologic function of *BcCSP3* and lay the foundation for a deeper understanding of *BcCSP3* in Pak-choi.

## 2. Materials and Methods

### 2.1. Experiment Material and Stresses

Seeds of Pak-choi (*Brassica rapa* ssp. *chinensis* cv. *Suzhouqing*) were obtained from the Cabbage System Biology Laboratory of Nanjing Agricultural University. The seeds were geminated and planted in a mixture of soil and sand (3:1, volume ratio). The seeds were then placed in an artificial climate room (16 h/8 h; 22/18 °C) and allowed to grow. After four weeks, the seedlings were exposed to various abiotic stress treatments including ABA, cold, salt and dehydration. For ABA, salt and dehydration treatments, 100 uM ABA, 200-mM NaCl and 15% (*w/v*) polyethylene glycol (PEG) was used, respectively [17]. For cold treatment, the seedlings were placed in a 4 °C incubator. Leaf tissues were collected at 0, 12, 24 and 48 h after initial exposure and then immediately placed in liquid nitrogen and promptly stored in a −80 °C freezer for further testing. To measure the expression level of *BcCSP3* throughout the plant, samples were taken from the roots, stems and leaves.

### 2.2. Identification and Isolation of BcCSP3

According to the sequence of *Arabidopsis* CSP protein, blasting in the non-heading Chinese cabbage database. In addition, in the non-heading Chinese cabbage database by blasting with the *Arabidopsis* genome, we found 6 genes containing CSP domains in the *B. rapa* genome, named *BcCSP1*, *BcCSP2*, *BcCSP3*, *BcCSP4*, *BcCSP5* and *BcCSP6* based on their sequence length [18]. Here, the focus of this work was *BcCSP3*. *BcCSP3* gene was identified and analyzed, then the primer design was performed using the method of comparing nucleotide sequences with the software Primer 5.0 (Appendix A). Total RNA was extracted from roots, stems and leaves with the RNAeasy mini kit (Tiangen, Beijing, China) following the manufacturer’s instructions. The cDNA of *BcCSP3* was amplified from a cDNA library of Pak-choi treated with diverse abiotic stresses by RT-qPCR. The first-strand cDNA was synthesized with 1 ug of mixed total RNA to construct a stress-induced cDNA library of Pak-choi by using a Superscript II kit (Takara, Dalian, China). Conserved regions were identified based on sequence information from the *Arabidopsis CSP* gene family (http://arabidopsis.org/index.jsp). The amplification program: 94 °C for 5 min, 30 cycles of 94 °C for 30 s, 60 °C for 30 s, 1 min at 72 °C, and for 10 min at 72 °C. The PCR product was cloned into pMD18-T (Takara, Dalian, China). The ORF size of *BcCSP3* was confirmed by sequencing. The molecular weight, isoelectric point (pI) and length of the putative protein were predicted using the online ExPasy program (http://www.expasy.org/tools/).

### 2.3. Bioinformatics Analysis of BcCSP3 Protein

Secondary structure and tertiary structure analysis were performed by online tool SOPMA [19] (https://npsa-prabi.ibcp.fr/cgi-bin/npsa_automat.pl?page=/NPSA/npsa_sopma.html) and SWISS-MODEL (https://www.swissmodel.expasy.org/), respectively; signal peptide prediction through the online tool SignalP-5.0 Server [20] (http://www.cbs.dtu.dk/services/SignalP); transmembrane regions prediction through the online tool TMpred [21] (http://www.ch.embnet.org/software/TMPRED_form.html); hydrophilic prediction through the online tool ProtScale [22] (https://web.expasy.org/protscale/).

### 2.4. Sequence Alignment and Phylogenetic Tree

Some CSP amino acid sequences from *Arabidopsis* were also obtained (Appendix A). The deduced amino acid sequence was analyzed using DNAMAN (Lynnon Biosoft, San Ramon, CA, USA). Multiple sequence alignment was performed by ClustalW [23] and MEGA 6.0 software [24] with the default parameters and manual correction. Phylogenetic trees were generated with MEGA 6.0 using the maximum likelihood method. Bootstrap values were estimated with 1000 replicates. The conserved motifs were analyzed using MEME suite (http://meme.nbcr.net/meme/) with the default settings except the maximum width was set to 200, and the maximum and minimum numbers of motifs were defined as 10 and 2, respectively.

### 2.5. Analysis of BcCSP3 under Stress Treatments

The roots, stems and leaves of non-heading Chinese cabbage were sampled separately, to study organ-specific expression analysis. To investigate the changes of *BcCSP3* expression under diverse stress treatments, we performed RT-qPCR. Real-time quantitative PCR is a commonly used high-throughput technique to measure mRNA transcription levels [25]. Total RNA was extracted from the previously frozen plant tissues with an RNA extraction kit (RNAsimply total RNA Kit, Tiangen, Beijing, China). Genomic DNA contamination was removed by using DNase I (Takara, Dalian, China). The first-strand cDNA was synthesized by using the PrimeScript™ RT reagent Kit (Takara, Dalian, China). The RT-qPCR assay was carried out with the 7500 fast real-time qPCR System (Applied Biosystems, Foster City, CA, USA). The PCR program: 95 °C for 5 min, followed by 40 cycles of 95 °C for 30 s, 60 °C for 30 s and 72 °C for 30 s. The gene-specific and internal control primers are listed in Appendix A. The relative expression level of *BcCSP3* was calculated by the 2^−ΔΔCT^ method [26]. All of the above RT-qPCR experiments were performed with three biologic replicates and three technical replicates.

### 2.6. Subcellular Localization of BcCSP3 Protein

The WoLF PSORT (https://wolfpsort.hgc.jp/) online tool was used to predict the subcellular localization of the BcCSP3 protein. The gateway-specific primers (Appendix A) were used to amplify the full-length coding region of BcCSP3 then construct the cloned product into the corresponding vector. The vector was then introduced into pEarleyGate103 using the gateway system (Invitrogen, Carlsbad, CA, USA) to get the *BcCSP3–GFP* fusion construct protein. In order to achieve transient expression of BcCSP3, *BcCSP3–GFP* was bombarded into tobacco leaves cells. The infected tobacco was grown in the dark at 22 °C for approximately 48 h and then placed under a confocal laser scanning microscope to find the corresponding fluorescence (Zeiss, Jena, Germany).

### 2.7. Protein–Protein Interaction Analysis

As a protein interaction prediction method, protein–protein interaction (PPI) refers to two (or more) protein molecules forming a protein complex through non-covalent bonds [27]. In biologic processes, protein–protein interactions and recognition play a certain role [28]. In order to predict the interaction between proteins and build a protein interaction network based on the prediction results, the STRING online tool (https://string-db.org/) was used.

## 3. Results

### 3.1. Identification and Cloning of BcCSP3

The full-length of *BcCSP3* gene was isolated. *BcCSP3* has a 618 bp ORF, encoding 205 deduced aa (Figure 1), with molecular weight of 19.39 kDa, pI of 5.92, the grand average of hydropathicity (GRAVY) of −0.764 and an instability index of 59.39 (Table 1). The ORF was confirmed by three repeated sequencing events.

### 3.2. Analysis of Proteins

SOPMA online tool was often used to predict the secondary structure of *BcCSP3* protein, with result shows that the secondary structure consists of 0.0293 alpha helix, 0.2439 extended strand, 0.2341 beta turn and 0.4927 random coil. Random coil, extended strand and beta turn are scattered throughout the protein (Figure 2). In addition, the predicted tertiary structure of the protein was consistent with the predicted results of its secondary structure (Figure 3). SignalP-5.0 was used to predict the signal peptide, and the results showed that: Signal Peptide (Sec/SPI), 0.0017%; Other (the probability that the sequence does not have any kind of signal peptide), 0.9983%, which means that *BcCSP3* has no signal peptide (Appendix A). Using TMHMM to predict the transmembrane domain of *BcCSP3* the result shows no transmembrane domain (Appendix A). ProtScale was used to analyze the hydrophilicity of *BcCSP3* protein, it is a hydrophilic protein (Appendix A).

### 3.3. Sequence Alignment and Phylogenetic Analysis

Using DNAMAN (Lynnon Biosoft, San Ramon, CA, USA) software, the deduced amino acid sequence of BcCSP3 was compared with the CSP protein sequence of *Arabidopsis*. It was found that the protein sequence of BcCSP3 had a certain similarity with AtCSP, especially a higher sequence similarity was detected at the N-terminus. Sequence alignment revealed that the BcCSP3 and AtCSP2 proteins had approximately 75.11% amino acid sequence homology. It is interesting that five CSPs all contain a well-preserved CSD field (Figure 4).

In order to reveal the evolutionary genetic relationship between BcCSP3 and AtCSPs, amino acid sequence alignment was used to establish a phylogenetic tree by setting 1000 bootstrap repeats with maximum likelihood method (Figure 5). The results showed that BcCSP3 had the closest genetic relationship with AtCSP2 in *Arabidopsis*. At the same time, after analyzing the motifs, it is interesting to find that the motifs of BcCSP3 and AtCSP2 are also very similar (both have 4 motif sites) (Figure 5).

### 3.4. BcCSP3 Expression Pattern under Various Abiotic Stresses

Through RT-qPCR analysis, we found that the expression level of *BcCSP3* in stems and leaves was much higher than that in roots (Figure 6). In addition, the relative expression of *BcCSP3* under abiotic stress was tested by RT-qPCR. As shown in Figure 7A, under the ABA treatment, the expression level of *BcCSP3* exhibited a tendency to increase first, then decrease before increasing again within 48 h. In the cold treatment, the expression level of *BcCSP3* kept increasing over time within 48 h (Figure 7B). Under the salt stress treatment, the expression level of *BcCSP3* increased first and then decreased and reached the maximum expression level at 24 h (Figure 7C). Under the dehydration stress, the expression trend of *BcCSP3* was basically similar to that expressed during the ABA treatment (Figure 7D). After cold treatment, the expression level of *BcCSP3* increased with time, indicating that cold treatment could affect expression level of *BcCSP3*.

### 3.5. Subcellular Localization Analysis

Through WoLF PSORT, *BcCSP3* was predicted to be located in the nucleus, and its nuclear localization score was 14.0 (KNN = 14). The subcellular localization of *BcCSP3* was tested in tobacco leaf cells by expressing transiently the *BcCSP3–GFP*. Results indicate the *BcCSP3–GFP* localized in the nucleus and cytoplasm (Figure 8).

### 3.6. Protein–Protein Interaction Network

By using the online tool STRING, we constructed a protein interaction network (Figure 9). From the selected database, it was found that ten proteins may have some relationship with BcCSP3 (Appendix A). Seven proteins (except Bra040811.1-P, Bra032735.1-P and Bra013111.1-P) were identified to interact with BcCSP3 through previous experimentally determined; further research found that Bra040811.1-P and Bra032735.1-P proteins were identified as having gene co-occurrence relationship; while nine proteins (except Bra013111.1-P) had a co-expression relationship with BcCSP3.

## 4. Discussion

*CSP* genes have been reported in many plants, including *Arabidopsis*, rice and wheat [8]. It is well documented that cold stress often affects the normal growth and development of plants. Studies have revealed that CSP is a protein associated with cold stress [29]. In this study, *BcCSP3* gene was isolated from Pak-choi by homologous sequence alignment analysis. Further research shows that the *BcCSP3* protein has an N-terminal CSD domain. The sequence alignment of *BcCSP3* and the *Arabidopsis* CSP family proteins indicates that they have a highly conserved N-terminus (CSD domain). Based on the above results, we speculate that *BcCSP3* has similar functions as *AtCSP* (Figure 4). The phylogenetic analysis shows that *BcCSP3* and *AtCSP2* are orthologous genes, so we predict that these two genes may have been conserved through evolution and possibly come from the same ancestor (Figure 5).

A large number of studies have shown that plant *CSP* genes can actively respond to abiotic stresses (especially cold stress) from the surrounding environment during growth. For example, *AtCSP3* [13] and *AtCSP1* [12] can exhibit a response to cold stress. Although overexpressed *AtCSP1* did not show a significant increase in frost resistance, *AtCSP1* was able to protect against cold damage by saving cold-sensitive grp7 mutants [16]. *AtCSP3* knock out plants showed more sensitivity to cold stress [12], while plants that overexpressed *AtCSP3* showed higher tolerance to cold stress [13]. Because *BcCSP3* and AtCSPs have similarities in amino acid sequence structure, we speculate that *BcCSP3* has similar functions and may also participate in the cold stress response.

Previous studies have found that drought, salt and ABA stress treatments show up-regulated expression level of *AtCSP3* [30]. Here, RT-qPCR analysis showed that the expression pattern of *BcCSP3* is different from *AtCSPs* in some respects. *BcCSP3* showed a positive response to cold, ABA and salt stress treatment while previous studies have shown that *AtCSP1* exhibits a down-regulated response under the drought and salt stresses [16]. Further research found that under the dehydration and ABA stress treatment, *BcCSP3* will be induced to show similar expression patterns and response trends (Figure 7). These results indicate that *BcCSP3* may not behave exactly the same under various stresses, compared with *AtCSPs*.

Previous research findings indicate that the subcellular localization of AtCSP1 [31] and AtCSP3 [13] are both located in the nucleus. AtCSP1 shows selective binding to the corresponding mRNA and can continue to maintain translation in response to various stresses [31]. A further study found that AtCSP3 also plays an important role in RNA processing and metabolism [30]. Here, we used the transient expression of *BcCSP3–GFP* fusion protein in tobacco leaf cells to show that BcCSP3 is located in the nucleus and cytoplasm (Figure 8). Based on the similarity of the amino acid sequences of *AtCSP1*, *AtCSP3* and *BcCSP3*, their conservative structure under the cold stress and similar subcellular localization, we speculate that it is very likely that the *BcCSP3* protein plays the role of RNA chaperone during cold adaptation.

## 5. Conclusions

In conclusion, a protein related to cold stress was identified and isolated from non-heading Chinese cabbage, BcCSP3, which is a new member of the Pak-choi CSP family and has the closest evolutionary relationship with AtCSP2 in *Arabidopsis*. BcCSP3 is a hydrophilicity cold-related protein with no signal peptide and no transmembrane domain. The expression level of *BcCSP3* in stems and leaves are higher than in roots of Pak-choi. At the same time, *BcCSP3* also shows response to other abiotic stresses and participates in active regulation. This study may help to further provide ideas for the mining of genes related to cold stress in non-heading Chinese cabbage.

## Figures and Tables

**Figure 1 plants-09-00890-f001:**
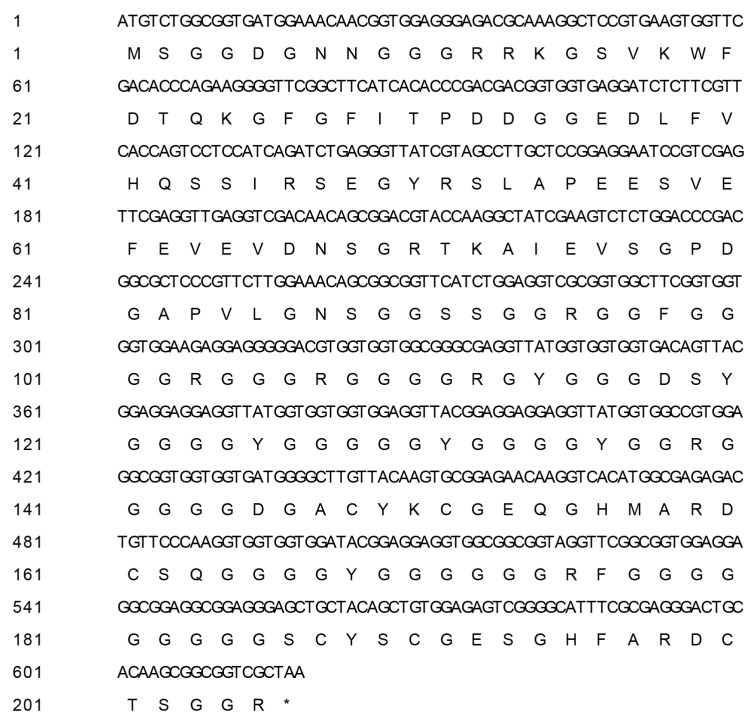
Nucleotide acid and corresponding amino acid sequence of BcCSP3 in non-heading Chinese cabbage.

**Figure 2 plants-09-00890-f002:**
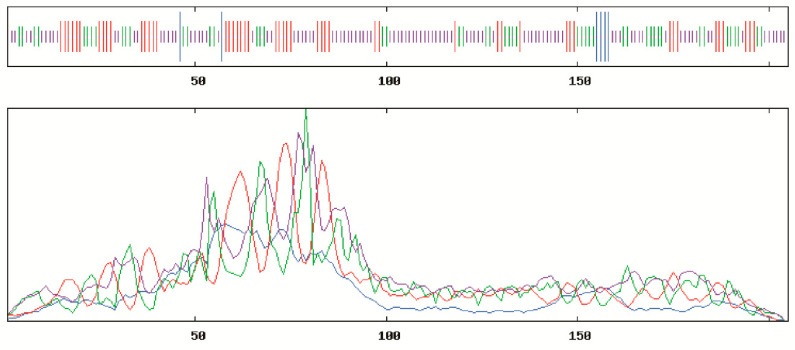
Secondary structural prediction of the BcCSP3 protein. The successively shorter straight lines represent helix, turn, strand and coil, respectively; the blue, red, green and purple curves represent helix, turn, strand and coil, respectively.

**Figure 3 plants-09-00890-f003:**
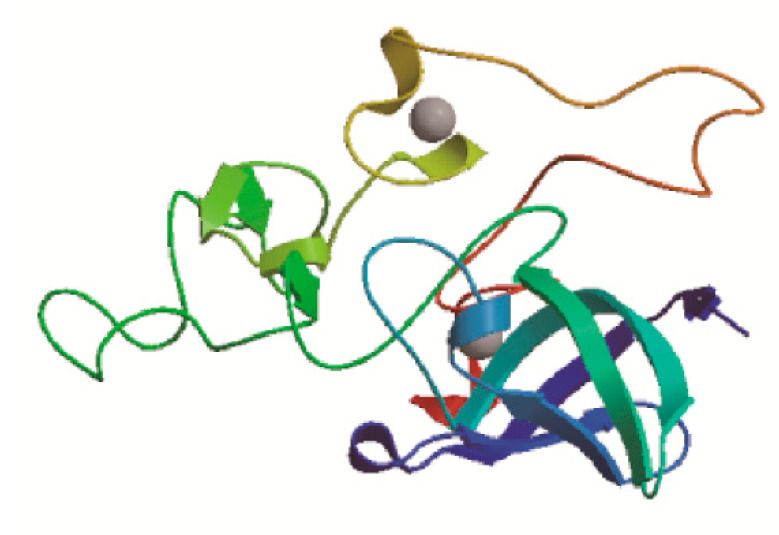
Tertiary structure prediction of the BcCSP3 protein. Random coil, extended strand and beta turn make up most of the main body.

**Figure 4 plants-09-00890-f004:**
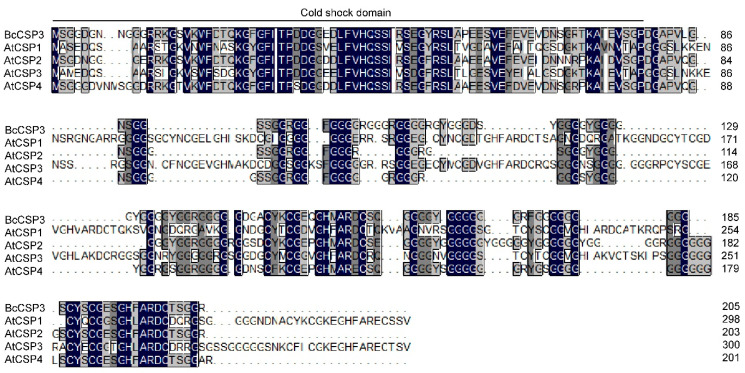
Amino acid sequence alignment of the putative BcCSP3 protein with *Arabidopsis* CSP proteins. Perfectly matched residues, highly conserved residues, and less conserved residues are represented by a black, dark gray and gray box, respectively.

**Figure 5 plants-09-00890-f005:**
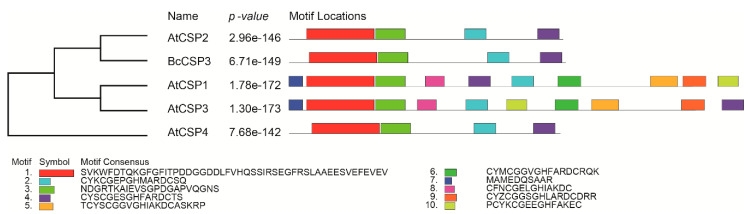
Phylogenetic and motif analysis of the putative BcCSP3 protein and *Arabidopsis* CSPs. The unrooted tree is based on an alignment of the full-length protein sequences and was constructed by the neighbor joining method.

**Figure 6 plants-09-00890-f006:**
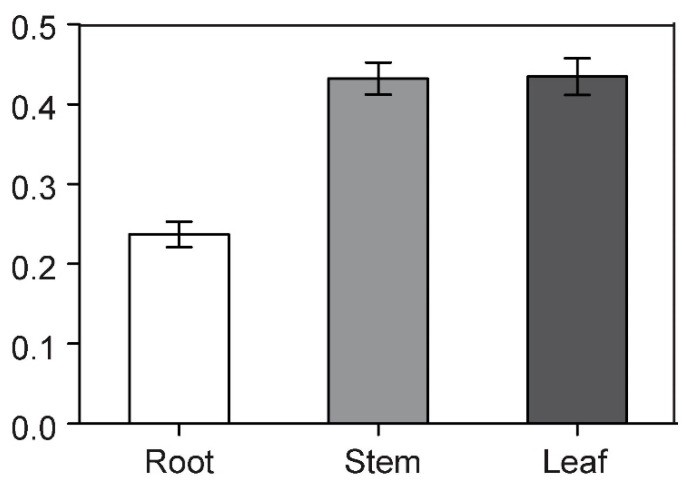
RT-qPCR analysis of the expression patterns of *BcCSP3* in roots, stems and leaves of Pak-choi. Each column represents the average and standard deviation of three replicates. Each bar value represents the mean ± SD (*p* < 0.05).

**Figure 7 plants-09-00890-f007:**
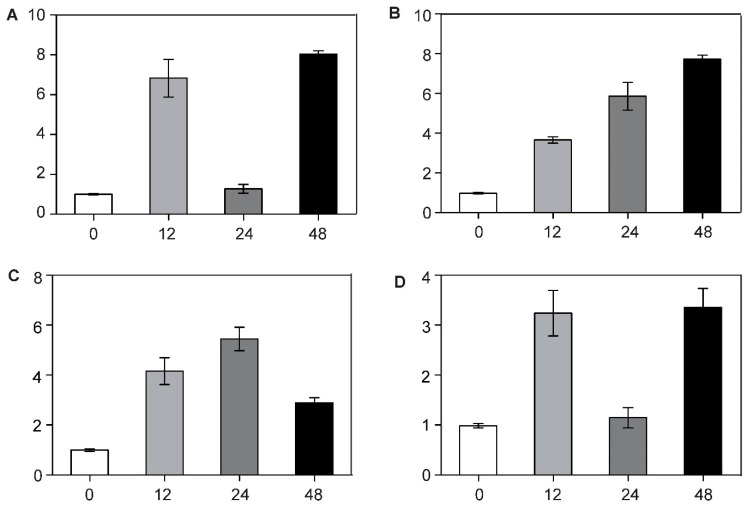
RT-qPCR analysis of the expression patterns of *BcCSP3* in Pak-choi (leaves) under abiotic stresses. (**A**) *BcCSP3* expression under ABA treatment; (**B**) *BcCSP3* expression under cold treatment; (**C**) *BcCSP3* expression under salt treatment; (**D**) *BcCSP3* expression under dehydration treatment. Each column represents the average and standard deviation of three replicates. Each bar value represents the mean ± SD (*p* < 0.05).

**Figure 8 plants-09-00890-f008:**
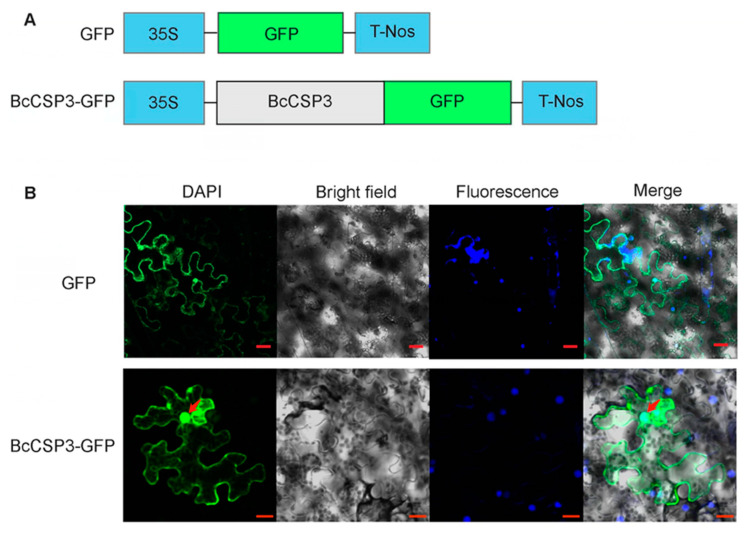
Subcellular localization of BcCSP3. (**A**) Constructs used in the experiments; (**B**) BcCSP3–GFP was transferred to tobacco leaf cells to test and verify the subcellular localization of BcCSP3 (bar = 20 μm).

**Figure 9 plants-09-00890-f009:**
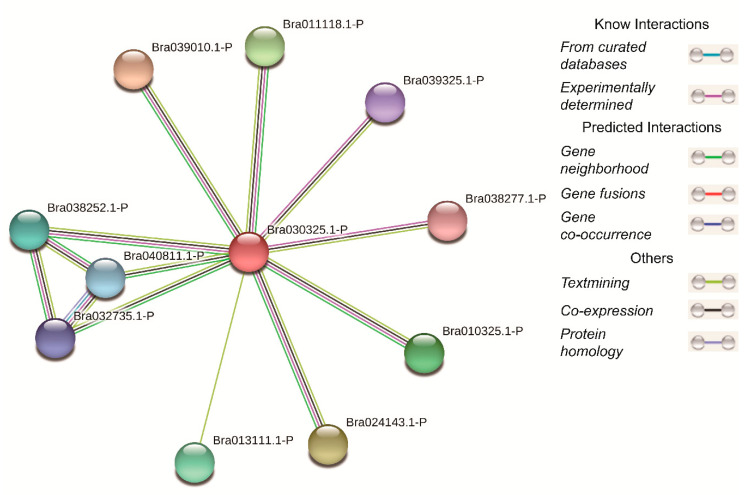
Online prediction of protein interactions.

**Table 1 plants-09-00890-t001:** General information on cold shock proteins (CSPs).

Protein Name	ORF (bp)	Amino Acid	WM (kDa)	pI	GRAVY	Instability Index
BcCSP3	618	205	19.39	5.92	−0.764	59.39
AtCSP1	900	299	30.09	8.18	−0.687	33.62
AtCSP2	612	203	19.15	5.62	−0.790	55.39
AtCSP3	906	301	29.56	7.41	−0.521	53.95
AtCSP4	606	201	19.08	6.29	−0.693	63.08

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
