# Peer review of "Identification and Expression Analysis of Cold Shock Protein 3 (BcCSP3) in Non-Heading Chinese Cabbage (Brassica rapa ssp. chinensis)"

_plants, 2020, doi:10.3390/plants9070890_

Round 1

Reviewer 1 Report

I`d like to ask the authors to accept the following suggestions and revise the content in the manuscript as much as they can. 

1)Authors have provided the important information in the letter to reviewer. The information is the following: 

By using the 4 CSP genes reported in Arabidopsis thaliana and performing blast sequence alignment in the non-heading Chinese cabbage database, 6 BcCSP genes were obtained, which were divided into: BcCSP1 to BcCSP6 according to their sequence length. According to the sequence analysis, it is predicted that BcCSP3 may participate in the cold stress response, so BcCSP3 is tested and analyzed to see if there is such a response relationship.  

Authors should add  the information in the manuscript also.  Additionally, authors should describe why they focus on only BcCSP3, but not others BcCSP1,2,4~6 there.  So, the manuscript will be logically understandable.

(For example, was only BcCSP3 among 6 BcCSP genes the most similar to Arabidopsis CSP genes? Such information would be important. Otherwise,   readers still will wonder why authors focused on BcCSP3.) 

2) Please cite(re-submit) the figure 8 containing the red arrows in the manuscript. 

3)In Result 3.6 protein-protein interaction network:

Although authors state they can provide only the name of the interacting proteins in the letter, they are actually not the names. 

What are  `Bra039010`, `Bra038252`,`Bra11118` and so on? At least, they should describe what each is in the manuscript. Otherwise, the current result will be meaningless. 

For example, I am not sure though, Bra039010 may be histone-arginine N-methyltransferase .... (I found it from google) 

In the discussion part, author can also include the sentences regarding the future work plan for studying the interaction of BcCSP3 with those proteins.

4)line 55. there is `CBF`. Authors should explain what it is or state its original name (C-repeat (CRT)/dehydration responsive element (DRE)-binding factor (CBF) with its reference in the manuscript

5) If possible, please describe briefly the phenotype of plants during the treatment for figure 7 in the manuscirpt. Are the treated stresses weak or strong for plants?

6)I can often find inappropriate English expressin in the manuscript. Please recheck overall. Especially, discussion part. 

Examples are following: I`d like to note that they are not all. So please ask   native English speaker to check your manuscript. 

  • line 253, we speculatle that whether   ---> please, omit `whether`,  then just state` we speculate BcCSP3 may have similar function...`or `We suppose BcCSP3 may~`
  • line 265 : also the same case with above
  • line 270-271: Previous studies have found that AtCSP1 will exhibits --> Please remove `will`. 
  • line 277: please remove `will`
  • line 230-231: BcCSP3 subcellular localization of tobacco leaf cells transiently expressing the BcCSP3-GFP fusion construct was tested.

       ==> In this sentence, the subject is too long. please revise it. : The subcellular localiztion of BcCSP3 was tested using tobacco leaf cells transiently expressing the BcCSP3-GFP. 

or  The subcellular localization of BcCSP3 was tested in tobacco leaf cells by  expressing transiently the BcCSP3-GFP

  • line 208-209: The expression level of BcCSP3 under different abiotic stresses was detected by RT-qPCR. 

       --> was measured    (I don`t think level is detected) 

  • line 172-173 

SignalP-5.0, which was used to predict the signal peptide, and the results showed

--> Please remove ` , which`. 

or --> The results from SignalP 5.0 for predicting the signal peptide showed

  • line 56: AtCSP2 has the function of negatively regulating freezing tolerance. 

--> AtCSP2 plays a role as the negative regulator of freezing tolerance. 

or--> AtCSP2 negatively regulates freezing tolerance.

Author Response

Reviewer 1

Dear, Thank you again for your comments.

Comments and Suggestions for Authors

1) Authors have provided the important information in the letter to reviewer. The information is the following:

By using the 4 CSP genes reported in Arabidopsis thaliana and performing blast sequence alignment in the non-heading Chinese cabbage database, 6 BcCSP genes were obtained, which were divided into: BcCSP1 to BcCSP6 according to their sequence length. According to the sequence analysis, it is predicted that BcCSP3 may participate in the cold stress response, so BcCSP3 is tested and analyzed to see if there is such a response relationship. Authors should add the information in the manuscript also. Additionally, authors should describe why they focus on only BcCSP3, but not others BcCSP1,2,4~6 there. So, the manuscript will be logically understandable.

(For example, was only BcCSP3 among 6 BcCSP genes the most similar to Arabidopsis CSP genes? Such information would be important. Otherwise, readers still will wonder why authors focused on BcCSP3.) 

Reply: --- Thank you for your comments and reminding. We have previously analyzed all the genes of the CSP gene family in non-heading Chinese cabbage and found that CSP3 is more sensitive to cold stress. Besides, we analyzed the protein sequence of CSP3, the CSP3 protein sequence contains a more obvious cold response domain. Also because Arabidopsis COLD SHOCK DOMAIN PROTEIN2 is a RNA chaperone that is regulated by cold and developmental signals (Sasaki K et al., 2007). BcCSP3 among 6 BcCSP genes the most similar to Arabidopsis CSP2 gene. Through sequence alignment, BcCSP3 and AtCSP2 protein have a high similarity, and through evolutionary tree analysis, BcCSP3 and AtCSP2 are more closely related. So we speculate whether they have similar cold stress emergency response. In addition, we added some content in the Introduction, please see revised manuscript (line 70-line 72). Therefore, we analyzed CSP3 to see if it has such a cold response. Thank you!

2) Please cite (re-submit) the figure 8 containing the red arrows in the manuscript.

Reply: --- Thank you for your comments and reminding. We have re-submitted the figure 8 containing the red arrows in the manuscript, please see new figure 8 (with red arrows). Thank you.

3) In Result 3.6 protein-protein interaction network: Although authors state they can provide only the name of the interacting proteins in the letter, they are actually not the names. What are `Bra039010`, `Bra038252`,`Bra11118` and so on? At least, they should describe what each is in the manuscript. Otherwise, the current result will be meaningless.

For example, I am not sure though, Bra039010 may be histone-arginine N-methyltransferase .... (I found it from google). In the discussion part, author can also include the sentences regarding the future work plan for studying the interaction of BcCSP3 with those proteins.

Reply: --- Thank you for your kind comments and reminding. We have added a supplementary table S3 to illustrate more information about these proteins, please see Supplementary Table S3. Besides, We have also added some content about Figure 9, Please see revised manuscript (line 218-line 224). We sincerely hope you can accept them. Thank you.

4) line 55. there is `CBF`. Authors should explain what it is or state its original name (C-repeat (CRT)/dehydration responsive element (DRE)-binding factor (CBF) with its reference in the manuscript.

Reply: --- Thank you for your kind reminding. We have added more descriptions, please see revised manuscript (line 59-line 60). Thank you.

5) If possible, please describe briefly the phenotype of plants during the treatment for figure 7 in the manuscirpt. Are the treated stresses weak or strong for plants?

Reply: --- Thank you for your comments and suggestions. We are so sorry that we do not provide the plant phenotype changes at that time. But when we experimented before, we didn't pay special attention to the phenotype of plants (treated stresses). According to common sense, there should be phenotypic changes. Thank you very much for your understanding and support. Thank you.

6) I can often find inappropriate English expressin in the manuscript. Please recheck overall. Especially, discussion part. Examples are following: I`d like to note that they are not all. So please ask native English speaker to check your manuscript.

line 253, we speculatle that whethe. ---> please, omit `whether`, then just state` we speculate BcCSP3 may have similar function...`or `We suppose BcCSP3 may~`

line 265 : also the same case with above

line 270-271: Previous studies have found that AtCSP1 will exhibits --> Please remove `will`.

line 277: please remove `will`

Reply: --- Thank you for your kind comments and reminding. We have modified these sentences and misspelled words. Please see revised manuscript (line 233-line 234, line 245, line 250-line 251, line 257-line 258). In addition, we also invited an American to modify the language of our manuscript. Thank you.

line 230-231: BcCSP3 subcellular localization of tobacco leaf cells transiently expressing the BcCSP3-GFP fusion construct was tested. ==> In this sentence, the subject is too long. please revise it. : The subcellular localiztion of BcCSP3 was tested using tobacco leaf cells transiently expressing the BcCSP3-GFP. Or The subcellular localization of BcCSP3 was tested in tobacco leaf cells by expressing transiently the BcCSP3-GFP.

line 208-209: The expression level of BcCSP3 under different abiotic stresses was detected by RT-qPCR. --> was measured (I don`t think level is detected)

Reply: --- Thank you for your comments and reminding. We have modified these wrong sentences, please see revised manuscript (line 213-line 214, line 200-line 201). In addition, we invited an American to modify the language of our manuscript. Thank you.

line 172-173: SignalP-5.0, which was used to predict the signal peptide, and the results showed. --> Please remove ` , which`. or --> The results from SignalP 5.0 for predicting the signal peptide showed.

line 56: AtCSP2 has the function of negatively regulating freezing tolerance. --> AtCSP2 plays a role as the negative regulator of freezing tolerance. or--> AtCSP2 negatively regulates freezing tolerance.

Reply: --- Thank you for your comments. You have very good suggestions for us, we have modified these sentences, please see revised manuscript (line 176-line 177, line 61-line 62). In addition, we invited an American to modify the language of our manuscript. Thank you.

Dear, Thank you again for your comments. All the changes in the text are highlighted in yellow, please check, we sincerely hope that these modifications will meet your requirements, thank you very much. Stay safe, best regards!

Reviewer 2 Report

1. Based on response from authors, the B. rapa genome has 6 genes containing CSP domains named BcCSP1, BcCSP2, BcCSP3, BcCSP4, BcCSP5 and BcCSP6. Still I could not understand why they chose BcCSP3 only for further analysis. I strongly suggest authors to analyze all BcCSPs (expression pattern, subcellular localization and so on)
2. If table 1 and 2 are not new findings, please move to supplementary materials.
3. As I suggested earlier, subcellular-localization of BcCSP3-GFP has to confirm by co-localization with organelle markers. For example, usually, DAPI staining is used to visualize nuclei localization. In addition, GFP signal can be observed in nucleus and cytoplasm, similar with BcCSP3-GFP (Figure 8). I really wonder whether nucleus and cytoplasm localization of BcCSP3-GFP is due to GFP ?. Firstly, I strongly suggest authors to analyze BcCSP3-GFP protein using western blotting. Furthermore, labeling of fluorescence and bright field need to change.
4, The authors have presented only preliminary results (obtained by simple-database-mining), which are not sufficient to publish in Plants. At least, complementation analysis using Arabidopsis T-DNA mutants is needed.

Author Response

Reviewer 2

Dear, Thank you for your comments. Comments and Suggestions for Authors.

  1. Based on response from authors, the B. rapa genome has 6 genes containing CSP domains named BcCSP1, BcCSP2, BcCSP3, BcCSP4, BcCSP5 and BcCSP6. Still I could not understand why they chose BcCSP3 only for further analysis. I strongly suggest authors to analyze all BcCSPs (expression pattern, subcellular localization and so on)

Reply: --- Thank you for your comments and reminding. We have previously analyzed all the genes (expression pattern) of the CSP gene family in non-heading Chinese cabbage and found that CSP3 is more sensitive to cold stress. Besides, we analyzed the protein sequence of CSP3, the CSP3 protein sequence contains a more obvious cold response domain. Also because Arabidopsis COLD SHOCK DOMAIN PROTEIN2 is a RNA chaperone that is regulated by cold and developmental signals (Sasaki K et al., 2007). BcCSP3 among 6 BcCSP genes the most similar to Arabidopsis CSP2 gene. Through sequence alignment, BcCSP3 and AtCSP2 protein have a high similarity, and through evolutionary tree analysis, BcCSP3 and AtCSP2 are more closely related. So we speculate whether they have similar cold stress emergency response. In addition, we added some content in the Introduction, please see revised manuscript (line 70-line 72). Therefore, we analyzed CSP3 to see if it has such a cold response. Thank you.

  1. If table 1 and 2 are not new findings, please move to supplementary materials.

Reply: --- Thank you for your recommendation. The table 1 and 2 have been moved to supplementary materials. Please check the revised version (Supplementary Table S1 and Supplementary Table S2). Thank you for your suggestion.

  1. As I suggested earlier, subcellular-localization of BcCSP3-GFP has to confirm by co-localization with organelle markers. For example, usually, DAPI staining is used to visualize nuclei localization. In addition, GFP signal can be observed in nucleus and cytoplasm, similar with BcCSP3-GFP (Figure 8). I really wonder whether nucleus and cytoplasm localization of BcCSP3-GFP is due to GFP?. Firstly, I strongly suggest authors to analyze BcCSP3-GFP protein using western blotting. Furthermore, labeling of fluorescence and bright field need to change.

Reply: --- Thank you for your comments and reminding. We am so sorry and thank you. We have provided a new Figure 8, the direction indicated by the red arrow is the position of the nucleus, DAPI staining is used to visualize nuclei localization. Besides, labeling of fluorescence and bright field have been changed, please see the new Figure 8. We are so sorry for not providing this experiment, western blotting. In order to meet the graduation requirements (publish SCI), there is not enough time to provide this experiment, we hope to get your understanding, we am so sorry and thank you.

  1. The authors have presented only preliminary results (obtained by simple-database-mining), which are not sufficient to publish in Plants. At least, complementation analysis using Arabidopsis T-DNA mutants is needed.

Reply: --- Thank you for your comments and reminding. Thank you very much for your suggestion, and we agree. We also tried the genetic modification verification (CSP3) test, but we failed. In order to meet the graduation requirements this time, we hope to publish this article to meet the graduation requirements. In addition, we need to complete this major revisions within the specified time (10 days). Thank you very much and agree with your suggestions. We hope to get your encouragement and support, thank you very much. Thank you.

Dear, Thank you again for your comments. All the changes in the text are highlighted in yellow, please check, we sincerely hope that these modifications will meet your requirements, thank you very much. Stay safe, best regards!

Reviewer 3 Report

The manuscript by Huang et al. identified a gene encoding BcCSP3 in non-heading Chinese cabbage, and analyzed stress-responsive expression patterns of BcCSP3. As the authors concluded in the abstract that this work might provide a reference for the identification and expression analysis of other CSP genes in non-heading Chinese cabbage, it will be interesting to identify and analyze the function of BcCSPs in cold stress response. Considering that approximately four CSPs are found in different plant species, identification of all CSPs in non-heading Chinese cabbage and analysis of their comprehensive stress-responsive expression patterns will provide much deeper insight into their roles in stress response.

Author Response

Reviewer 3

Dear, Thank you for your comments.

Comments and Suggestions for Authors

The manuscript by Huang et al. identified a gene encoding BcCSP3 in non-heading Chinese cabbage, and analyzed stress-responsive expression patterns of BcCSP3. As the authors concluded in the abstract that this work might provide a reference for the identification and expression analysis of other CSP genes in non-heading Chinese cabbage, it will be interesting to identify and analyze the function of BcCSPs in cold stress response. Considering that approximately four CSPs are found in different plant species, identification of all CSPs in non-heading Chinese cabbage and analysis of their comprehensive stress-responsive expression patterns will provide much deeper insight into their roles in stress response.

Reply: --- Thank you for your comments and friendly encouragement. Thank you. Best regards. Thank you again for your time and effort. Thank you very much. Stay safe, best regards!

Round 2

Reviewer 2 Report

I am really sorry, but I could not satisfied that they have addressed my previous comments.

Author Response

Dear reviewer,

Hello, dear reviewer. Thank you very much for your careful review (Minor revisions) of our manuscript (Article Manuscript-ID: plants-838685). All authors have completed the modification of the article and submitted it in order to hope to meet your requirements. We deeply appreciate your consideration of our manuscript, we hope this paper is suitable for ‘Plants-Basel (Plants)’. Thanks a lot for your invaluable comments which helped very much to improve the quality of the manuscript. According to the requirements, we have completed this Minor revisions within the prescribed time (3 days). We have made the following revision. Thank you for your valuable and thoughtful comments. Thanks!

Based on response from authors, the B. rapa genome has 6 genes containing CSP domains named BcCSP1, BcCSP2, BcCSP3, BcCSP4, BcCSP5 and BcCSP6. Still I could not understand why they chose BcCSP3 only for further analysis. I strongly suggest authors to analyze all BcCSPs (expression pattern, subcellular localization and so on)

Reply: --- Thank you for your comments and reminding and we agree. In previous work, we have identified all CSPs in untitled Chinese cabbage and analyzed their stress response expression patterns. This time, we added a description in this regard and inserted a reference (perhaps at the beginning of section 2.2) (Huang F et al., 2019), please review the revised manuscript (line 95-99). We have previously analyzed all the genes (expression pattern) of the CSP gene family in non-heading Chinese cabbage. Besides, we analyzed the protein sequence of CSP3, the CSP3 protein sequence contains a more obvious cold response domain. Also because Arabidopsis COLD SHOCK DOMAIN PROTEIN2 is a RNA chaperone that is regulated by cold and developmental signals (Sasaki K et al., 2007). BcCSP3 among 6 BcCSP genes the most similar to Arabidopsis CSP2 gene. Through sequence alignment, BcCSP3 and AtCSP2 protein have a high similarity, and through evolutionary tree analysis, BcCSP3 and AtCSP2 are more closely related. So we speculate whether they have similar cold stress emergency response. In addition, we have to complete this minor revisions within the specified time (3 days). Thank you very much and agree with your suggestions. We hope to get your encouragement and support. Thank you.

Reviewer 3 Report

In my original review, I asked the authors to identify all CSPs in non-heading Chinese cabbage and analyze their comprehensive stress-responsive expression patterns to provide much deeper insight into the roles of BcCSPs in stress response. However, in this revision, no attempt was made to address this issue.

Author Response

Dear reviewer,

Hello, dear reviewer. Thank you very much for your careful review (Minor revisions) of our manuscript (Article Manuscript-ID: plants-838685). All authors have completed the modification of the article and submitted it in order to hope to meet your requirements. We deeply appreciate your consideration of our manuscript, we hope this paper is suitable for ‘Plants-Basel (Plants)’. Thanks a lot for your invaluable comments which helped very much to improve the quality of the manuscript. According to the requirements, we have completed this Minor revisions within the prescribed time (3 days). We have made the following revision. Thank you for your valuable and thoughtful comments. Thanks!

Comments and Suggestions for Authors

The manuscript by Huang et al. identified a gene encoding BcCSP3 in non-heading Chinese cabbage, and analyzed stress-responsive expression patterns of BcCSP3. As the authors concluded in the abstract that this work might provide a reference for the identification and expression analysis of other CSP genes in non-heading Chinese cabbage, it will be interesting to identify and analyze the function of BcCSPs in cold stress response. Considering that approximately four CSPs are found in different plant species, identification of all CSPs in non-heading Chinese cabbage and analysis of their comprehensive stress-responsive expression patterns will provide much deeper insight into their roles in stress response.

In my original review, I asked the authors to identify all CSPs in non-heading Chinese cabbage and analyze their comprehensive stress-responsive expression patterns to provide much deeper insight into the roles of BcCSPs in stress response. However, in this revision, no attempt was made to address this issue.

Reply: --- Thank you for your comments and friendly encouragement. In previous work, we have identified all CSPs in untitled Chinese cabbage and analyzed their stress response expression patterns. This time, we added a description in this regard and inserted a reference (perhaps at the beginning of section 2.2) (Huang F et al., 2019), please review the revised manuscript (line 95-99). We have previously analyzed all the genes (expression pattern) of the CSP gene family in non-heading Chinese cabbage. Besides, we analyzed the protein sequence of CSP3, the CSP3 protein sequence contains a more obvious cold response domain. Also because Arabidopsis COLD SHOCK DOMAIN PROTEIN2 is a RNA chaperone that is regulated by cold and developmental signals (Sasaki K et al., 2007). BcCSP3 among 6 BcCSP genes the most similar to Arabidopsis CSP2 gene. Through sequence alignment, BcCSP3 and AtCSP2 protein have a high similarity, and through evolutionary tree analysis, BcCSP3 and AtCSP2 are more closely related. So we speculate whether they have similar cold stress emergency response. In addition, we need to complete this minor revisions within the specified time (3 days). Thank you very much and agree with your suggestions. We hope to get your encouragement and support. Thank you. Thank you for your suggestion. Thank you. Best regards. Thank you very much. Stay safe, best regards!

This manuscript is a resubmission of an earlier submission. The following is a list of the peer review reports and author responses from that submission.

Round 1

Reviewer 1 Report

In this manuscript, Huang et al. identified cold shock protein 3 from non-heading Chinese cabbage (Brassica rapa ssp. chinensis cv. Suzhouqing), and analyzed its expression patterns under various stimuli. The data reported is important and provide a valuable resource to the community. However, since the completion of genome sequencing of B. rapa is available in the public database, it has been possible for the identification of gene families through the analysis of sequence similarity. Firstly, I would recommend the identification of CSP family in B. rapa based on genome-wide analysis. This analysis will provide more information such as evolutionary mechanisms of gene family. In addition, the experimental proof related with physiological function of BcCSP3, not only expression analysis, should be certainly expedite the manuscript.

1. How many CSP protein has been found in B. rapa genome? Why named as BcCPS3? Are there two more proteins?

2. BcCSP3 has high homology to Brara.D01243.1 (containing 'Cold-shock' DNA-binding domain) in B. rapa genome. But Brara.D01243.1 is 10 amino acids shorter than BcCSP3. Please provide figure for alignment of amino acid sequences of BcCSP3 and B.rapa cold shock protein member including Brara.D01243.1, and Figure 1 and 4 should be replaced by this figure. In addition, the authors have to discuss about sequence difference between BcCSP3 and B.rapa cold shock protein member

3. Information about Arabidopsis CSPs in table 2 and 3 is new finding ?

4. Figure 8

Please provide co-localization results with organelle markers

5. Figure 9.

Although authors have described that ten proteins were found to interact with BcCSP3, these proteins are AtCSP3-interacting proteins. Co-expression analysis of B.rapa CSP (homologues of BcCSP3) will provide more information rather than AtCSP3-interacting proteins.

6. I would suggest that the authors combine the result and discussion together, in only one section, since the discussion by itself is rather short, and they already include few “discussion like” sentences while describing the results. If authors decided to keep results and discussion separate, then they should move these sentences (basically, anything with a citation on it) to the Discussion section.

7. Identification of cold shock protein from non-heading Chinese cabbage is a good beginning, but not enough. I wondered why the authors did not analyze the functions of the genes deeply. For example, the functions of the genes could be further investigated by using transgenic seedlings. Thus, the result of this manuscript is preliminary.

Reviewer 2 Report

 In this study, authors isolated the gene for BcCSP3 from Pak-choi. They found that its mRNA is more expressed in both stem and leaf compared to root and that its expression is responsive to ABA, cold, salt and dehydration. Moreover, they reported its protein is located in the nucleus and cytoplasm. Additionally,  they performed and presented the data for silico analysis using the amino acid sequences.

Since the information on the genes in pak-choi is limitted, the subject itselfof this paper is considered to be quite attractive.However, the data generated from the experiments is very few and most of data is based on the silico analysis whose evidence is not actually confirmed. Furthermore, their explination is not sufficient, even though they show a lot of data from anlysis of amino acid sequences. At least, authors should demonstrate the following:

  1. Why is the gene name BcCSP3?  There is not any information on other BcCSP1 or 2 in the manuscript. Also, I wonder why author focused BcCSP3
  2. Althors stated taht the BcCSP3 gene was identified based on the coserved region of Arabidopsis CSP genes .  But the process is unclear in the method, because the primers(probably, BcCSP3-S ,A) are the sequence of the BcCSP itself. They should explain how they designed the primers using the compare the nucleotide sequenes in more detail.
  3. They analyzed the amino acids sequences by using web tool and then found several motifs. But they didn`t mention about them after that. I think authors should disscuss what those motifs mean for their funtion.
  4.  In protein-protein network, authors found the possibility that eight proteins may interact with BcCSP3. However, they don`t provide any information on those proteins. Please explain them.
  5. In the figure about the subcellular localization of BcCSP3, the locations of the nucleus seem unclear.